# A three-country analysis of the gut microbiome indicates taxon associations with diet vary by taxon resolution and population

Lora Khatib,[1,2] Se Jin Song,[3] Amanda H. Dilmore,[1,4] Jon G. Sanders,[3] Caitriona Brennan,[1,5] Alejandra Rios Hernandez,[1] Tyler Myers,[3] Renee Oles,[1] Sawyer Farmer,[1] Charles Cowart,[1,3] Amanda Birmingham,[1] Edgar A. Diaz,[1] Oliver Nizet,[6] Kat Gilbert,[1] Nicole Litwin,[7] Promi Das,[1,3] Brent Nowinski,[3] Mackenzie Bryant,[1] Caitlin Tribelhorn,[1] Karenina Sanders-Bodai,[1] Soline Chaumont,[8] Jan Knol,[9,10] Guus Roeselers,[9] Manolo Laiola,[9] Sudarshan A. Shetty,[9] Patrick Veiga,[8] Julien Tap,[8] Muriel Derrien,[8] Hana Koutnikova,[8] Aurélie Cotillard,[8] Christophe Lay,[11] Armando R. Tovar,[12] Nimbe Torres,[12] Liliana Arteaga,[12] Antonio González,[1] Daniel McDonald,[1] Andrew Bartko,[1,3,13] Rob Knight[1,3,13,14]

**ABSTRACT** Emerging research suggests that diet plays a vital role in shaping the composition and function of the gut microbiota. Although substantial efforts have been made to identify general patterns linking diet to the gut microbiome, much of this research has been concentrated on a small number of countries. Additionally, both diet and the gut microbiome have highly complex and individualized configurations, and there is growing evidence that tailoring diets to individual gut microbiota profiles may optimize the path toward improving or maintaining health and preventing disease. Using fecal metagenomic data from 1,177 individuals across three countries, we examine the relationship between diet and bacterial genera, focusing on *Prevotella* and *Faecalibacterium*, which have gained significant attention for their potential roles in human health and strong associations with dietary patterns. We find that these two genera in particular show significant associations with many aspects of diet but these associations vary in scale and direction, depending on the level of metagenomic resolution (i.e., genus level by reads and strain level by metagenome-assembled genomes) and the contextual population. These results highlight the growing importance of building metagenomic data sets that are standardized, comprehensive, and representative of diverse populations to increase our ability to tease apart the complex relationship between diet and the microbiome.

**IMPORTANCE** An analysis of fecal microbiome data from individuals in the United States, United Kingdom, and Mexico shows that associations with dietary components vary both by country and by level of resolution (i.e., genus and strain). Our work sheds light on why there may be conflicting reports regarding microbial associations with diet, disease, and health.

**KEYWORDS** human microbiome, metagenomics, diet, *Prevotella*, *Faecalibacterium*

In this study, we explored the relationships between the gut microbiome at different levels (taxonomic genera and metagenome-assembled genomes [MAGs]) and various dietary factors using data collected from subjects in three countries (Fig. 1a and b). We conducted metagenomic sequencing of fecal samples from 442 adult participants in the United States (US), 342 in the United Kingdom (UK), and 507 in Mexico, recruited through the Microsetta Initiative platform (1). We evaluated long-term dietary intake using the VioScreen food frequency questionnaire (FFQ; Version 5, VioCare, Princeton, NJ) (2), adapting it for Mexico to include common regional foods (Version 5-Mex). Consumption of food groups (kcal/day) was derived from raw VioScreen entries, applying several

**Peer Reviewers** José Francisco Cobo Díaz, Universidad de Leon, Leon, Spain; Lianmin Chen, University of Groningen, Groningen, the Netherlands

Address correspondence to Andrew Bartko, abartko@ucsd.edu.

R.K. is a scientific advisory board member and consultant for BiomeSense, Inc., has equity, and receives income. He is a scientific advisory board member and has equity in GenCirq. He is a consultant and scientific advisory board member for DayTwo and receives income. He has equity in and acts as a consultant for Cybele. He is a co-founder of Biota, Inc., and has equity. He is a co-founder of Micronoma, has equity, and is a scientific advisory board member. The terms of this arrangement have been reviewed and approved by the University of California, San Diego, in accordance with its conflict-of-interest policies. A.B. is a founder of Guilden Corporation and is an equity owner. The terms of these arrangements have been reviewed and approved by the University of California, San Diego, in accordance with its conflict-of-interest policies. D.M. is a consultant for BiomeSense, Inc., has equity, and receives income. The terms of these arrangements have been reviewed and approved by the University of California, San Diego in accordance with its conflict-of-interest policies. C.L., A.C., H.K., M.D., J.T., P.V., S.A.S., M.L., G.R., J.K., and S.C. are employees of Danone.

See the funding table on p. 6.

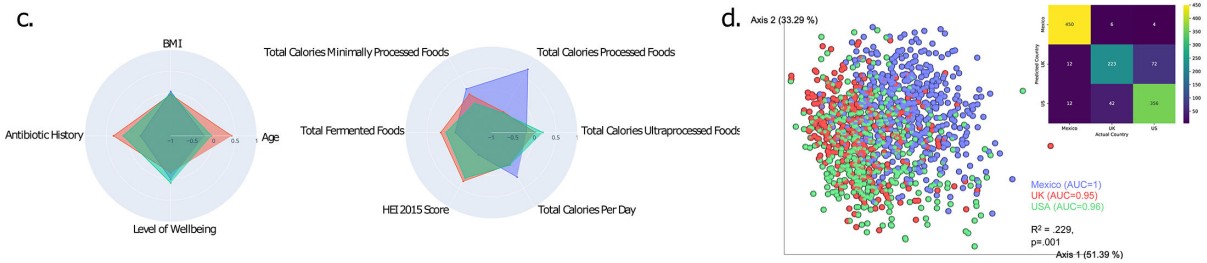

**FIG 1** Study and data overview. (a) The table shows the number of samples analyzed by shotgun metagenomics from each country and the resulting sample size after filtering steps. (b) Data collected included taxonomic profiles and MAGs from the sequence data, and diet and lifestyle information from self-reported answers to questionnaires. (c) Radar plots show *z*-normalized values for key personal and diet-related variables, averaged by country. (d) A principal coordinates analysis plot shows robust Principal Component Analysis (PCA) distance among the microbiome samples, colored by the cohort country. Partial R2 from a Permutational Multivariate Analysis of Variance (PERMANOVA) is reported. A confusion matrix shows the classification accuracy using a random forest classifier (fivefold cross validation) on the taxonomic feature table. The mean Area Under the Curve (AUC) for the Receiver Operating Characteristic (ROC) curve is also reported.

filtering criteria to ensure data quality based on the number of items consumed, total energy intake, and FFQ completion time as previously described (3).

All cohorts included more females than males, with an average body mass index (BMI) of 25 (±5 SD), but varied in other characteristics. Subjects in the UK were older, while those in Mexico reported higher incidences of diabetes and irritable bowel disease (IBD) and the lowest diet quality (Healthy Eating Index [HEI]; Fig. 1c; Table 1). However, all three cohorts constituted individuals healthier than the general US population, reporting a lower BMI, higher HEI, and lower incidence of diabetes and cardiovascular disease (4, 5).

Stool collection and DNA extraction were performed similarly across the three cohorts. Sample libraries generated an average of 4,559,420 sequences per sample (±1,750,993 SD), which were quality controlled and human sequence filtered following Armstrong et al. (6), before generating operational genomic units through the woltka pipeline (7) (Fig. 1b). A robust PCA analysis (8) showed that geographic location had the strongest effect on microbiome composition (partial $R^2$ = 0.229, $P$ = 0.001; Fig. 1d), enabling a random forest classifier to predict location with high accuracy (mean AUC: 1.00 for Mexico, 0.95 for UK, and 0.96 for USA) across fivefold cross-validation. The next top variables showing significant effects not related to diet were antibiotic history ($R^2$ = 0.0137, $P$ = 0.001), mental and physical well-being ($R^2$ = 0.0132, $P$ = 0.001), and BMI ($R^2$ = 0.0076, $P$ = 0.001). Therefore, all downstream analyses, where possible, included these factors as covariates.

To narrow the focus for downstream, strain-resolved analyses, we assembled contigs and derived species-level genome bins (sGBs) using dRep (9), following Sanders et al. (10). Next, we calculated nucleotide diversity within each sGB for each individual using InStrain (11) and then down-selected from 286 dietary variables using linear LASSO regression, followed by linear mixed-effects models. This investigation revealed that *Prevotella* (including some species now reclassified under different genera, e.g., *Segatella copri*) and *Faecalibacterium* nucleotide diversity showed the strongest associations with diet (Fig. 2a). This finding was consistent across countries (Fig. 2b) but varied across certain covariate subgroups—though *Prevotella* and *Faecalibacterium* consistently showed more associations than most genera (Fig. S1). *Prevotella* diversity across all participants was positively associated with whole grain consumption ($t$ = 4.962, $P$ = 7.21e−7) and negatively with antibiotic use ($t$ = −6.949, $P$ = 4.17e−12). *Faecalibacterium*

**TABLE 1** Key demographic, health, and diet characteristics of the three cohorts

| Parameter | Mexico | UK | US | Statistic | P-value |
|---|---|---|---|---|---|
| No. of participants | 460 (39.1%) | 307 (26.1%) | 410 (34.8%) | 31.026 | 1.83e−07 |
| Age | 42.195 ± 15.133 | 52.116 ± 13.269 | 45.011 ± 15.22 | 81.618 | 1.89e−18 |
| Sex: female | 307 (66.7%) | 223 (72.6%) | 262 (63.9%) | 6.191 | 0.04525496 |
| Education level: college graduate | 206 (45.8%) | 153 (50.2%) | 189 (46.8%) | 20.238 | 0.00044809 |
| Alcohol frequency: regularly | 36 (7.9%) | 80 (26.4%) | 92 (22.4%) | 52.158 | 4.72e−12 |
| BMI | 25.573 ± 5.01 | 25.091 ± 5.706 | 25.23 ± 5.314 | 5.114 | 0.0775445 |
| Diabetes | 31 (6.9%) | 9 (3.0%) | 15 (3.7%) | 7.777 | 0.02047483 |
| Cardiovascular disease (CVD) | 9 (2.0%) | 14 (4.6%) | 16 (3.9%) | 9.818 | 0.04360907 |
| Autoimmune disease | 19 (4.3%) | 51 (17.0%) | 52 (12.8%) | 34.029 | 4.08e−08 |
| Inflammatory bowel disease (IBD) | 64 (15.1%) | 9 (3.0%) | 9 (2.3%) | 61.016 | 5.63e−14 |
| Irritable bowel syndrome (IBS) | 123 (28.4%) | 82 (27.1%) | 72 (18.3%) | 12.9 | 0.00158089 |
| Gluten intolerance | 26 (6.3%) | 31 (10.3%) | 51 (12.9%) | 23.303 | 0.00070106 |
| Lactose intolerance | 136 (31.1%) | 29 (9.9%) | 74 (18.6%) | 49.665 | 1.64e−11 |
| Bowel movement, normal | 349 (77.0%) | 218 (73.4%) | 300 (76.5%) | 30.367 | 3.35e−05 |
| Diet type (vegetarian) | 6 (1.3%) | 21 (6.9%) | 27 (6.6%) | 83.655 | 8.96e−15 |
| Plant diversity (more than 20) | 55 (12.1%) | 105 (34.9%) | 96 (23.5%) | 189.206 | 1.20e−36 |
| Vegetable frequency, regularly | 358 (78.0%) | 286 (93.5%) | 345 (84.8%) | 37.794 | 1.24e−07 |
| Fruit frequency, regularly | 314 (68.6%) | 207 (67.9%) | 251 (61.5%) | 6.348 | 0.17461969 |
| Whole grain frequency, regularly | 194 (42.7%) | 155 (50.7%) | 243 (59.7%) | 25.219 | 4.55e−05 |
| Red meat frequency, regularly | 171 (37.6%) | 46 (15.0%) | 60 (14.7%) | 90.161 | 1.22e−18 |
| Milk/cheese frequency, regularly | 303 (66.0%) | 188 (61.2%) | 198 (48.9%) | 28.107 | 1.19e−05 |
| Sugar-sweetened beverage frequency, regularly | 46 (10.1%) | 7 (2.3%) | 13 (3.2%) | 76.907 | 7.87e−16 |
| Energy intake from the diet (kcal/day) | 2284.797 ± 901.074 | 2007.523 ± 688.092 | 2017.892 ± 678.689 | 21.896 | 1.76e−05 |
| Ultraprocessed foods (kcal/day) | 345.554 ± 127.168 | 438.919 ± 141.507 | 487.323 ± 152.632 | 195.972 | 2.79e−43 |
| Processed foods (kcal/day) | 213.998 ± 100.985 | 80.455 ± 62.177 | 77.713 ± 59.797 | 493.984 | 5.40e−108 |
| Minimally processed foods (kcal/day) | 388.19 ± 121.734 | 411.377 ± 130.741 | 359.726 ± 133.215 | 34.562 | 3.13e−08 |
| Carbohydrate (%) | 47.283 ± 7.625 | 46.105 ± 9.872 | 44.178 ± 9.259 | 28.152 | 7.71e−07 |
| Fat (%) | 34.088 ± 6.295 | 34.988 ± 8.698 | 36.854 ± 8.159 | 30.223 | 2.74e−07 |
| Protein (%) | 16.307 ± 3.189 | 14.753 ± 3.007 | 15.341 ± 3.526 | 47.901 | 3.97e−11 |
| HEI (2010) | 64.741 ± 9.495 | 73.197 ± 10.537 | 72.456 ± 10.516 | 182.008 | 3.00e−40 |
| HEI (2015) | 63.348 ± 9.308 | 71.027 ± 10.242 | 70.166 ± 10.398 | 147.01 | 1.19e−32 |

diversity was positively associated with dietary fiber ($t$ = 9.282, $P$ = 2.20e−20), consistent with previous studies (12), and, unexpectedly, with processed meat ($t$ = 9.060, $P$ = 1.67e−19)—possibly reflecting participants in Mexico who reported high fiber and processed meat consumption.

Next, we assessed the associations between the strain-level abundance of these two genera and a subset of dietary variables that broadly captured overall dietary patterns (Table S1). Only 14 out of 56 *Prevotella* sGBs showed significant associations with dietary variables (Fig. 2c). Few *Prevotella* sGBs showed consistency across the cohorts. In the full data set, *Prevotella disiens* was positively associated with alcohol consumption, *P. sp002481295* was associated with animal and protein intake, and two sGBs, *P. MAG sGB 00348* and *P. copri sGB 00022* (i.e., *Segatella copri*), were associated with starch intake. However, the remaining 10 sGBs showed associations that were country specific. For example, *P. timonensis* (i.e., *Hoylesella timonensis*) showed positive associations with vegetable and fiber intake in the US, but not in the UK or Mexico, where it was instead associated with saturated fat intake.

In contrast, 17 out of 19 of the *Faecalibacterium* sGBs showed significant associations (Fig. 2c). Several *Faecalibacterium* sGBs (most notably *F. MAG_sGB_00436*, *F. prausnitzii_I*, and *F. sp900539885*) were positively associated with intake of vegetable, fruit, carbohydrates (including fibers, pectin, and starch), and unprocessed or minimally processed foods, as well as an overall healthy diet (high HEI), while negatively associated with

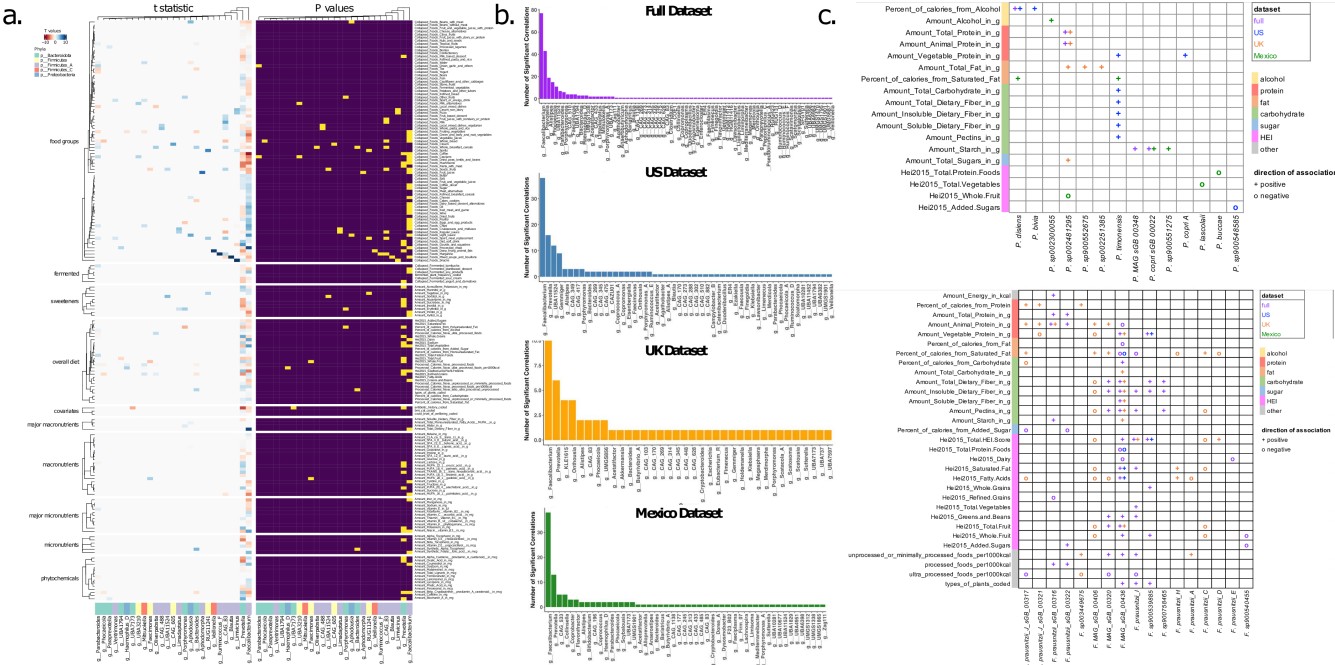

**FIG 2** Aspects of *Prevotella* and *Faecalibacterium* at different levels of resolution highlight the complexity of diet-microbe associations, each showing ties with many dietary variables, but varying in direction and strength by country. (a) Heatmap shows *t*-scores and *P*-values for testing the correlation between the nucleotide diversity in sGBs and dietary variables, pooled per individual by bacterial genus. (b) Bar plots show the number of significant correlations between each bacterial genus and dietary variables across all participants (purple), as well as within each country: US (blue), UK (orange), and Mexico (green). (c) An association map shows the dietary variables that were significantly correlated with the log ratio of each *Prevotella* sGB to the sum of all *Prevotella* (top) and each *Faecalibacterium* sGB to the sum of all *Faecalibacterium* (bottom) in the full data set and stratified by country, after accounting for the variation explained by covariates (cohort, BMI, antibiotic history, and level of well-being). Multiple comparisons were corrected using the Benjamini-Hochberg method with a 5% False Discovery Rate (FDR).

animal protein and dairy. However, other sGBs (*F. prausnitzii_C* and *F. MAG_sGB_00406*) showed opposite trends, being associated with animal protein, saturated fat, and an overall poorer diet (low HEI), but only in the UK cohort. Interestingly, no sGBs were found to be significantly associated with any dietary variables in Mexico.

Lastly, we examined whether the Centered Log-Ratio (CLR)-transformed values of *Prevotella* and *Faecalibacterium* displayed a similar pattern to nucleotide diversity. We found that, when analyzed on the genus level, only *Prevotella* exhibited a positive correlation with starch intake ($R^2 = 0.0317$, $P = 0.04$), and this association was only consistent in participants from Mexico (Fig. S2a). The absence of consistent genus-level abundance associations, in contrast to diversity and strain-level findings, emphasizes that resolving microbiome features to the strain level is essential for identifying biologically meaningful relationships with diet.

Understanding the complex associations between diet and the microbiome can contribute to our knowledge of gut microbiota modulation and its implications for personalized nutrition. Here, we focused on two bacterial genera, *Prevotella* and *Faecalibacterium*, which have shown the strongest associations with diet and human health across studies. Both taxa can ferment dietary fibers (12–14) and are associated with plant-based foods (15, 16). *Prevotella* has been associated with both beneficial and detrimental health effects. For example, while *Prevotella* is associated with high-fiber diets (14), it has also been associated with inflammatory conditions, such as periodontal disease, rheumatoid arthritis, and metabolic syndrome (17–19). *Faecalibacterium*, by contrast, has consistently been associated with beneficial health effects (20). The strain-level contributions to these associations are not yet fully understood, but growing studies have revealed that *Prevotella* and *Faecalibacterium* are richer in strain diversity

than previously appreciated (21, 22). A recent study showed that *Faecalibacterium* strain diversity can vary among people of different ages, populations, lifestyles, and disease statuses (23), with populations from less industrialized regions exhibiting higher prevalence and diversity. In our study, although individuals from Mexico did not exhibit markedly different *Prevotella* abundances at the genus level compared to those from the US (Fig. S2a), *Prevotella* strains displayed distinct, population-specific patterns (Fig. S2c). Conversely, while *Faecalibacterium* abundance varied across cohorts at the genus level (Fig. S2b), strain-level distributions were relatively consistent (Fig. S2d). These findings underscore the limitations of genus-level conclusions and highlight the importance of strain-level resolution when interpreting microbiome relationships.

Microbiome research has largely been limited to specific populations, which restricts generalizability. Our study takes a step toward addressing this gap; however, caution is warranted when interpreting findings from a minimal subset of the global population. Further research is needed to confirm these trends. Capturing a broader range of human and dietary diversity will help us untangle the complex relationships between microbial strains and diet. Our study indicates that specific associations with diet are detected only at the strain level. Efforts to develop a better understanding of whether other components of diet or the microbiome modify the relationships between a given strain and a given dietary item in a region-specific manner are warranted.

## ACKNOWLEDGMENTS

We would also like to thank Dan Hakim, Senyen Luo, Dominic Nguyen, Jeff DeReus, Cassidy Symons, Alex Richter, Brittanie Collinsworth, Franck Lejzerowics, Sandrine Miller-Montgomery, Gail Ackermann, George Armstrong, Greg Humphrey, Diana Gutierrez Lopez, Tara Schwartz, and Gibraan Rahman for their support and contributions to this project.

This work was funded by Danone Nutricia Research and the Center for Microbiome Innovation and supported by the Microsetta initiative.

## AUTHOR AFFILIATIONS

[1]Department of Pediatrics, University of California San Diego, La Jolla, California, USA

[2]Neurosciences Graduate Program, University of California San Diego, La Jolla, California, USA

[3]Center for Microbiome Innovation, University of California San Diego, La Jolla, California, USA

[4]Biomedical Sciences Program, University of California San Diego, La Jolla, California, USA

[5]Division of Biological Sciences, University of California San Diego, La Jolla, California, USA

[6]Department of Chemical and Biomolecular Engineering, Johns Hopkins University, Baltimore, Maryland, USA

[7]Crohn's and Colitis Foundation, New York, New York, USA

[8]Danone Research and Innovation, Gif sur Yvette, France

[9]Danone Research and Innovation, Utrecht, the Netherlands

[10]Laboratory of Microbiology, Wageningen University and Research, Wageningen, the Netherlands

[11]Precision Nutrition D-Lab, Danone Global Research and Innovation Center, Singapore, Singapore

[12]Instituto Nacional de Ciencias Médicas y Nutrición Salvador Zubirán, Mexico City, México

[13]Department of Bioengineering, University of California San Diego, La Jolla, California, USA

[14]Department of Computer Science and Engineering, University of California San Diego, La Jolla, California, USA

## AUTHOR ORCIDs

Lora Khatib  http://orcid.org/0000-0002-6998-8278
Caitriona Brennan  http://orcid.org/0000-0003-3943-6701
Renee Oles  http://orcid.org/0000-0001-5945-0215
Promi Das  http://orcid.org/0000-0001-6733-2059
Andrew Bartko  http://orcid.org/0000-0002-1237-2747
Rob Knight  http://orcid.org/0000-0002-0975-9019

## FUNDING

| Funder | Grant(s) | Author(s) |
| --- | --- | --- |
| Danone Nutricia Research | | Andrew Bartko |
| | | Rob Knight |
| Center for Microbiome Innovation | | Andrew Bartko |
| | | Rob Knight |
| Microsetta Initative | | Andrew Bartko |
| | | Rob Knight |

## DATA AVAILABILITY

The data used in this study will be made available through the European Bioinformatics Institute (EBI) database (accession number: PRJEB11419) and Qiita Study #10317.

## ADDITIONAL FILES

The following material is available online.

### Supplemental Material

**Supplemental material (mSystems00544-25-s0001.docx).** Supplemental methods, Table S1, and Figures S1 and S2.

### Open Peer Review

**PEER REVIEW HISTORY (review-history.pdf).** An accounting of the reviewer comments and feedback.

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
