## [Reviewer comments · mSystems]

A three-country analysis of the gut microbiome indicates taxon associations with diet vary by taxon resolution and population

Lora Khatib, Se Jin Song, Amanda Dilmore, Jon Sanders, Caitriona Brennan, Alejandra Hernández, Tyler Myers, Renee Oles, Sawyer Farmer, Charles Cowart, Amanda Birmingham, Edgar Diaz, Oliver Nizet, Kat Gilbert, Nicole Litwin, Promi Das, Brent Nowinski, MacKenzie Bryant, Caitlin Tribelhorn, Karenina Sanders-Bodai, Soline Chaumont, Jan Knol, Guus Roeselers, Manolo Laiola, Sudarshan Shetty, Patrick Veiga, Julien Tap, Muriel Derrien, Hana Koutnikova, Aurélie Cotillard, Christophe Lay, Armando Tovar, Nimbe Torres, Liliana Arteaga, Antonio González, Daniel McDonald, Andrew Bartko, and Rob Knight

Corresponding Author(s): Andrew Bartko, University of California, San Diego

Review Timeline:

Submission Date:

April 15, 2025

Accepted:

June 4, 2025

Editor: Daniel Garrido

Reviewer(s): Disclosure of reviewer identity is with reference to reviewer comments included in decision letter(s). The following individuals involved in review of your submission have agreed to reveal their identity: José Francisco Cobo Díaz (Reviewer #1); Lianmin Chen (Reviewer #5)

Transaction Report:

DOI: <https://doi.org/10.1128/msystems.00544-25>

Re: mSystems00544-25 (A three-country analysis of the gut microbiome indicates taxon associations with diet vary by taxon resolution and population)

Dear Dr. Andrew Bartko:

Your manuscript has been accepted, and I am forwarding it to the ASM production staff for publication. Your paper will first be checked to make sure all elements meet the technical requirements. ASM staff will contact you if anything needs to be revised before copyediting and production can begin. Otherwise, you will be notified when your proofs are ready to be viewed.

Sincerely,
Daniel Garrido

Reviewer #4 (Comments for the Author):

This paper explores the relationships between the gut microbiome at different levels of resolution (genus and strains from metagenome-assembled genomes (MAGs)) and various dietary factors using data collected from individuals in three countries. The authors conducted metagenomic sequencing of fecal samples from 1,177 subjects in the US, UK, and Mexico. The work focuses mainly on two genera, *Prevotella* and *Faecalibacterium*.

The topic is interesting and tries to cover a gap in the microbiome research as previous works were limited to specific populations. One of the results of this study finds is that specific associations with diet can be detected only at the strain level. The results of this work help to optimize the path toward improving or maintaining health and preventing disease, especially that some diseases are due to nutritional problems.

The authors used appropriate protocols for data collection, extraction, and sequencing. They also implemented their computational analyses using sound computational approaches and appropriate software that are published in the literature for metagenomic data processing, MAGs assembly, binning, and annotation. They also chose the right methods to perform their statistical analyses.

The authors addressed all the reviewer's comments from previous review by conducting sensitivity analysis, adding the method of lambda selection, modifying the text to make it clearer, and adding a missing figure.

The results are presented clearly and sufficiently inside the text and in figures, tables, and supplemental materials, which makes the paper easy to read and understand. The interesting results and the knowledge contribution of this paper will be useful for researchers in the field.

A few minor issues:

- 1) On line 178 of the Supplemental Methods, the word "significant" should be "significantly".
- 2) On line 179 of the Supplemental Methods, the word "comparison" should be "comparisons".
- 3) On the reference list of the Supplemental Methods, the number 32 is used for two different references on lines 289 and 295.

Reviewer #5 (Comments for the Author):

The revised manuscript is in good shape; I have no further comments.